# Preparation of Functional Nanoparticles-Loaded Magnetic Carbon Nanohorn Nanocomposites towards Composite Treatment

**DOI:** 10.3390/nano13050839

**Published:** 2023-02-23

**Authors:** Fitriani Jati Rahmania, Yi-Shou Huang, Yitayal Admassu Workie, Toyoko Imae, Anna Kondo, Yukiko Miki, Ritsuko Imai, Takashi Nagai, Hiroshi Nakagawa, Noriyasu Kawai, Kaname Tsutsumiuchi

**Affiliations:** 1Graduate Institute of Applied Science and Technology, National Taiwan University of Science and Technology, Taipei 10607, Taiwan; 2Department of Chemical Engineering, National Taiwan University of Science and Technology, Taipei 10607, Taiwan; 3College of Bioscience and Biotechnology, Chubu University, 1200 Matsumoto, Kasugai 487-8501, Japan; 4Department of Nephron-Urology, Graduate School of Medical Sciences, Nagoya City University, Nagoya 467-8601, Japan

**Keywords:** carbon nanohorn, iron oxide, gadolinium oxide, samarium oxide, polyethylene glycol, drug co-delivery, photothermal effect, hyperthermia

## Abstract

Combination therapy for cancer is expected for the synergetic effect of different treatments, and the development of promising carrier materials is demanded for new therapeutics. In this study, nanocomposites including functional nanoparticles (NPs) such as samarium oxide NP for radiotherapy and gadolinium oxide NP as a magnetic resonance imaging agent were synthesized and chemically combined with iron oxide NP-embedded or carbon dot-coating iron oxide NP-embedded carbon nanohorn carriers, where iron oxide NP is a hyperthermia reagent and carbon dot exerts effects on photodynamic/photothermal treatments. These nanocomposites exerted potential for delivery of anticancer drugs (doxorubicin, gemcitabine, and camptothecin) even after being coated with poly(ethylene glycol). The co-delivery of these anticancer drugs played better drug-release efficacy than the independent drug delivery, and the thermal and photothermal procedures enlarged the drug release. Thus, the prepared nanocomposites can be expected as materials to develop advanced medication for combination treatment.

## 1. Introduction

Cancer is globally the second leading cause of death that is predicted to become the primary reason for death and the biggest obstacle to extending life in the following decades [1]. Although conventional cancer treatments such as surgery, chemotherapy, and radiation therapy have been successful to some extent, they still damage normal tissue and causes severe side effects [2,3]. Therefore, theranostics that integrate therapy and medical imaging should be developed to overcome unwanted bio-distribution variations and improve therapeutic efficacy [4].

Iron oxide NP (Fe_3_O_4_) nanoparticle (NP) has gained significant attention in the treatment of cancer, which is primarily used as hyperthermia and negative contrast agent in magnetic resonance imaging (MRI) [5,6]. Carbon dots (Cdot)-coating Fe_3_O_4_ (Fe_3_O_4_@Cdot) provided an oxidation barrier and prevented the degradation of magnetic core components, as well as the particle interactions [7]. Moreover, it enhances photothermal/photodynamic therapy (PTT/PDT) efficacy through its rapid generation dynamics of singlet oxygen [8,9]. Gadolinium-containing NPs are now under development due to the huge number of potential applications and considerable advancement of techniques and procedures for producing monodisperse and size-controllable nanoparticles. Especially gadolinium oxide (Gd_2_O_3_) acts as a positive MRI contrast agent [10] in contrast to a negative contrast agent Fe_3_O_4_. It could be simultaneously combined with Fe_3_O_4_ to improve T_1_ and T_2_ relaxation time [11]. Samarium-153 (^153^Sm) is an excellent beta-emitter radiotherapy with a short half-life (1.93 days) for treating malignant tumors, such as lung, bone, and prostate cancer [12,13,14]. ^153^Sm could be produced by neutron activation of sTable 1^53^Sm-enriched samarium oxide (Sm_2_O_3_) [15]. Moreover, it is helpful for imaging and enhancing the efficacy and safety of the treatment in lower radioactivity [16].

Among carbon nanomaterials, carbon nanohorn (CNH) has been used as a carrier and other biological applications due to its excellent advantages, such as homogenous morphology, low toxicity, and easy functionalization [17,18,19,20]. Because it is a nanostructure consisting of horn-shaped tips of graphitic sp^2^ carbon atoms and having 80–100 nm size, it also can accumulate into cancer cells through either passive (by enhanced permeability and retention effect) or active (by surface modification with a specific ligand) targeting [17]. Then, as well as graphene, graphene oxides and carbon nanotubes [21,22,23], CNH is also valuable for cancer therapy: phototheranostics has been achieved based on CNH [24], and the simultaneous imaging and drug delivery effects have been reported with functionalized CNH [25].

In the present study, three kinds of metal oxide NPs that may act for radiotherapy (Sm_2_O_3_), imaging (Gd_2_O_3_), and thermotherapy (Fe_3_O_4_) were synthesized to bind on CNH. Photothermal/photodynamic Cdot-coated Fe_3_O_4_ (Fe_3_O_4_@Cdot) NPs were also bound onto CNH instead of Fe_3_O_4_ NPs. Nanocomposites (CNH/Fe_3_O_4_/Sm_2_O_3_/Gd_2_O_3_ and CNH/Fe_3_O_4_@Cdot/Sm_2_O_3_/Gd_2_O_3_) were covalently bound with poly(ethylene glycol) (PEG), which improves the dispersibility of CNH carrier in water and the biocompatibility of CNH carrier in the human body [26,27]. Then the products can be useful for theranostics, e.g., imaging, chemotherapy, radiotherapy, thermotherapy, and photothermal/photodynamic therapies. Here, the efficiencies of the products for chemical, thermal and photothermal therapies are examined. Three cancer drugs were applied for chemotherapy. Doxorubicin (DOX) and gemcitabine (GEM) were used water-soluble drugs. Additionally, the delivery of hydrophobic camptothecin (CPT) and hydrophilic GEM and co-delivery of both drugs and, as a comparison, hydrophobic camptothecin (CPT) and hydrophilic GEM were also examined. Moreover, the thermal and photothermal effects on release of dual-drug were proved using a heater or magnet and light emitted diode (LED). The investigations demonstrate the performance of pH-responsive, synergic, and sustainable drug release, which has potential as a multi-delivery on cancer therapy and represents a step forward in developing advanced medication for combination therapy and theranostics. The cancer therapy using mainly operation, chemotherapy and radiotherapy is now on the crossway to develop advanced therapies to reduce the load on patients: one of them is a combination therapy, where multiple functional materials for different therapies are loaded on a carrier. Such reports are still less, except a report where Fe_3_O_4_, Sm_2_O_3_, and Gd_2_O_3_ NPs were encapsulated in a polydopamine carrier [28].

## 2. Experimental Section

### 2.1. Materials

Oxidized CNH was purchased from NEC, Japan. Samarium(III) nitrate hexahydrate (Sm(NO_3_)_3_∙6H_2_O, 99.9%), gadolinium(III) nitrate hexahydrate (Gd(NO_3_)_3_∙6H_2_O, 99.9%), and diethylene glycol (DEG, C_4_H_10_O_3_, 99%), poly(ethylene glycol) (PEG_400_), 4-dimethylaminopyridine (DMAP), N,N’-dicyclohexylcarbodiimide (DCC), dimethylformamide (DMF), and camptothecin (CPT, 98%) were purchased from Acros Organics (Geel, Belgium). Doxorubicin hydrochloride (DOX, >98%), gemcitabine (GEM, 98%), and regenerated cellulose tubular membrane (MWCO of 6000–8000 g mol^−1^) were products from Sigma-Aldrich (St. Louis, MO, USA), AK Scientific (Union City, CA, USA), and Sigma-Aldrich (St. Louis, MO, USA), Sigma-Aldrich (St. Louis, MO, USA), respectively. Other reagents are commercial grade.

### 2.2. Instruments

Fourier-transform infrared absorption spectra (FTIR) were measured on a NICOLET 6700, Thermo Scientific, Waltham, MA, USA, for KBr pellets. Images of transmission electron microscope (TEM, JEOL JEM–2100, 120 kV, Mitaka, Japan) and high-resolution TEM (HRTEM, FEI Tecnai™ G2 F-20 S-TWIN, Hillsboro, OR, USA) were analyzed with analysis software (Gatan Digital Micrograph 3.5). A dispersion was dropped on a carbon-coated copper grid and air-dried. A scanning electron microscope (SEM, JSM–6390, Tokyo, Japan) was equipped with an energy dispersive X-ray spectrometer (EDS) at 15 kV. Thermogravimetric analysis (TGA, TA Q500, New Castle, DE, USA) was performed under airflow at 20–800 °C with a heating rate of 20 °C/min. A dynamic light scattering analyzer (DLS ELS-Z, Photal Osaka Electronics, Osaka, Japan) was used to measure hydrodynamic diameters and zeta-potentials. Ultraviolet (UV)–visible absorption spectra were measured on a JASCO V-670, Tokyo, Japan.

### 2.3. Preparation of Materials

Sm_2_O_3_ and Gd_2_O_3_ NPs were synthesized based on the polyol solvent method previously reported [28,29]. Briefly, a solution of Sm(NO_3_)_3_∙6H_2_O (0.25 mmol) in DEG (9 mL) was adjusted to pH 11.5 using a DEG solution of NaOH (0.125 M). The mixture was refluxed at 140 °C for 1 h and 180 °C for 1 h until changing to a brown dispersion. After cooling, the supernatant was collected by centrifugation (6000 rpm, 30 min). The same procedure for Gd(NO_3_)_3_∙6H_2_O (0.2 M, 10 mL) in DEG was conducted to prepare Gd_2_O_3_ NPs. CNH, Fe_3_O_4_, Fe_3_O_4_@Cdot, CNH/Fe_3_O_4,_ and CNH/Fe_3_O_4_@Cdot are the same materials as previously synthesized and characterized [30]. CNH was produced by acid-treating commercial oxidized CNH with nitric acid under refluxing at 100 °C. Fe_3_O_4_ NPs was synthesized using the co-precipitation method [31], and Fe_3_O_4_@Cdot was prepared based on the hydrothermal method [32]. CNH/Fe_3_O_4_ nanocomposites were prepared via the in situ co-precipitation of Fe_3_O_4_ on CNH, and CNH/Fe_3_O_4_@Cdot was obtained by the amide bonding of Fe_3_O_4_@Cdot with CNH [30]. The characterization of these nanoparticles/nanocomposites was performed using FTIR, X-Ray Diffraction (XRD), TEM, SEM, X-ray photoelectron spectroscopic (XPS), TGA, and DLS [28,30]. Measurements for Fe_3_O_4_ and its nanocomposites proved the characteristics of Fe_3_O_4_ including a super-paramagnetic nature.

Preparation of nanocomposites of CNH, including Sm_2_O_3_ and Gd_2_O_3_, was performed using an esterification reaction [33,34]. DCC (10 mM, 80 µL) and DMAP (10 mM, 20 µL) were added to a CNH (10 mg) dispersion in DMF (10 mL). Sm_2_O_3_ (15 mg) and Gd_2_O_3_ (15 mg), separately, were added to the mixture and stirred for 24 h at room temperature (~25 °C). Obtained composites were collected, washed, and dried. CNH/Fe_3_O_4_/Sm_2_O_3_/Gd_2_O_3_ and CNH/Fe_3_O_4_@Cdot/Sm_2_O_3_/Gd_2_O_3_ were formed by esterification of CNH/Fe_3_O_4_ and CNH/Fe_3_O_4_@Cdot with Sm_2_O_3_ and Gd_2_O_3_ using the same procedure described for CNH/Sm_2_O_3_ and CNH/Gd_2_O_3_. In these processes, carboxylic acid on CNH or Cdot is esterified with DEG adsorbed on Sm_2_O_3_ and Gd_2_O_3_.

Functionalization of CNH/Fe_3_O_4_/Sm_2_O_3_/Gd_2_O_3_ and CNH/Fe_3_O_4_@Cdot/Sm_2_O_3_/Gd_2_O_3_ by covalent binding of PEG_400_ was performed by dissolving nanocomposites (20 mg) in DMF (10 mL), adding DCC (10 mM, 80 µL) and DMAP (10 mM, 20 µL), dropping PEG_400_ (30 mg) and keeping at room temperature for 72 h. The products were collected, washed, and dried. Separately, CNH coated with PEG was prepared similarly. The synthesis processes of CNH nanocomposites simultaneously embedded three metal oxides were illustrated in Figure 1.

### 2.4. Drug Loading and Release

The drug loading on carriers was conducted as in previous reports [30,35]. Drugs at different concentrations (10–110 μg/mL) were mixed with an aqueous dispersion of carrier (100 μg/mL) and stirred for 24 h in the dark. The dispersion was centrifuged (6000 rpm, 3 h), and the supernatant was collected to determine the amount of unbound drug from the absorbance at 268 and 480 nm for GEM and DOX, respectively. Loaded drugs on carriers were evaluated by subtracting unbound drugs from initial loaded drugs and divided by the amount of carrier [36]. The CPT loading was similarly performed using an absorption band at 360 nm in methanol because it is water insoluble.

The drug release investigation was carried out at a physiological temperature (37 °C) and a cell-killing temperature (42 °C). A dispersion of drug-loaded carrier in a dialysis membrane tube was dialyzed against an external phosphate buffer saline (PBS) solution (15 mL) at pH 5.5 and 7.4. The drug release under the photothermal condition was performed using LED blue light (450 nm, 0.08 Wcm^−2^) [35]. Release media (2 mL) was withdrawn and replenished with an equal volume of fresh media at various time intervals. The released drugs were quantified using the same colorimetric method as a case of drug loading and calculated as a % released amount divided by the loaded amount (*n* = 3) [35].

### 2.5. Viability Test

Crystal violet assay was used to determine the viability of NCI-N87 cells with CNH/Fe_3_O_4_@Cdot, CPT@CNH/Fe_3_O_4_@Cdot, and GEM&CPT@CNH/Fe_3_O_4_@Cdot delivery systems. First, 4 × 10^4^ cells of NCI-N87 were seeded with 100 μL medium in a 96-well plate and incubated at 37 °C under 5% CO_2_. After 72 h, a dispersion (100 μL) of delivery system was added to each well with 2-fold serial dilutions. After additional 24 h incubation, the medium and cells that undergo cell death were removed. All wells were washed twice with 100 μL of PBS. Then, 50 μL of 0.1% PBS solution of crystal violet was poured into each well. Fifteen (15) min later, the solution was removed. After washing with 100 μL of distilled water twice, the plate was dried at room temperature for 24 h. One hundred (100) μL of methanol was added and mixed well to dissolve the remaining crystal violet dye. Finally, the optical density of each well was measured at 570 nm with a plate reader.

## 3. Results and Discussion

### 3.1. Characterization of Materials

CNH/Gd_2_O_3_, CNH/Sm_2_O_3_, CNH/Fe_3_O_4_/Sm_2_O_3_/Gd_2_O_3_, CNH/Fe_3_O_4_@Cdot/Sm_2_O_3_/Gd_2_O_3_, PEG@CNH/Fe_3_O_4_/Sm_2_O_3_/Gd_2_O_3_, and PEG@CNH/Fe_3_O_4_@Cdot/Sm_2_O_3_/Gd_2_O_3_ nanocomposites were synthesized and characterized. Figure 1A shows IR absorption spectra of DEG, Sm_2_O_3_, and Gd_2_O_3_. The IR bands of DEG at 3399 and 1650 cm^−1^ correspond to the O-H stretching and bending vibration modes of hydroxyl group, and bands at 2943 and 2873 cm^−1^ can be assigned to CH_2_ stretching vibration modes of ethylene group. A C-O-C stretching vibration band of alkoxy group and a C-O(H) stretching vibration band of hydroxyl group appeared at 1128 and 1064 cm^−1^, respectively. The IR absorption spectra of Sm_2_O_3_ and Gd_2_O_3_ NPs at 4000–1000 cm^−1^ were utterly similar to those of DEG. These results indicate that DEG molecules adsorb on Sm_2_O_3_ and Gd_2_O_3_ NPs, and act as a surface capping agent to protect NPs [28,37].

An IR broad band of CNH ranging from 3600 to 3200 cm^−1^ is ascribed to O-H stretching vibration mode, and a band at 1705 cm^−1^ is C=O stretching mode of carboxylic acid group after the acid treatment. Absorption bands at 1577, 1387, and 1176 cm^−1^ can be assigned to C=C stretching, C-H bending, and C-O-C stretching modes [37]. Then, an IR spectrum of CNH/Gd_2_O_3_ is associated with vibration modes of CNH, and a spectrum of CNH/Sm_2_O_3_ is contributed with DEG an CNH.

Figure 1B(a,b) show TEM images of CNH/Gd_2_O_3_ and CNH/Sm_2_O_3_. There were no significant changes in the shape of CNH after combining with Gd_2_O_3_ and Sm_2_O_3_ NPs. However, small particles are found in CNH. Figure 1C(a,b) show HRTEM images of CNH/Sm_2_O_3_ and CNH/Gd_2_O_3_. Since the sizes of small particles in CNH were approximately 5 and 4 nm, similar to the previous report [28,38], which can be assigned to be Gd_2_O_3_ and Sm_2_O_3_ NPs, respectively, as Fe_3_O_4_ NPs in CNH/Fe_3_O_4_ [30]. These particle sizes of Sm_2_O_3_ and Gd_2_O_3_ NPs synthesized from the polyol solvent method were much smaller than those (~90 nm) of the calcination method [28]. The d-spacings of lattice fringes of Sm_2_O_3_ NPs were calculated as 0.27 nm and 0.32 nm corresponding to (400) and (222) planes, respectively, of cubic crystal structure [28,39]. In comparison, a calculated d-spacing of Gd_2_O_3_ NPs was 0.32 nm corresponding to a (222) crystal plane of cubic crystal structure [28,38]. Thus, the deposition of Sm_2_O_3_ and Gd_2_O_3_ NPs was confirmed on CNH, although the existence of these NPs was not necessarily confirmed from IR spectra.

When both Sm_2_O_3_ and Gd_2_O_3_ NPs were combined on CNH/Fe_3_O_4_, an IR spectrum (Figure 2A) of nanocomposites displayed a similar spectrum to that of CNH and DEG on Sm_2_O_3_ and Gd_2_O_3_ NPs because Fe_3_O_4_, Sm_2_O_3_, and Gd_2_O_3_ NPs do not have characteristic IR bands at wavenumber above 1000 cm^−1^ (Figure 1A) [30]. However, an IR spectrum (Figure 2A) of CNH/Fe_3_O_4_@Cdot/Sm_2_O_3_/Gd_2_O_3_ was different from it of CNH/Fe_3_O_4_/Sm_2_O_3_/Gd_2_O_3_. Because a spectrum is prevailed by bands of CNH/Fe_3_O_4_@Cdot: CNH/Fe_3_O_4_@Cdot has similar IR bands at 3391, 1645, 1569, 1401, 1146 cm^−1^, which are assigned to characteristic vibration modes of carboxylic acid, carboxylate and graphitic groups of CNH and Cdot [30].

The thermal behaviors of CNH/Fe_3_O_4_/Sm_2_O_3_/Gd_2_O_3_ and CNH/Fe_3_O_4_@Cdot/Sm_2_O_3_/Gd_2_O_3_ nanocomposites were investigated with TGA, as shown in Figure 2B. The first weight loss (7.5 and 6.0 wt%, respectively) at a temperature below 100 °C is due to removals of water and residual solvent. The weight losses of 27.0% at 350~580 °C for CNH/Fe_3_O_4_/Sm_2_O_3_/Gd_2_O_3_ and 18.6% at 450~570 °C for CNH/Fe_3_O_4_@Cdot/Sm_2_O_3_/Gd_2_O_3_ come from the thermal decomposition of CNH and organic protectors on metal oxides [32,40]. The remaining residues of metal oxides in CNH/Fe_3_O_4_/Sm_2_O_3_/Gd_2_O_3_ and CNH/Fe_3_O_4_@Cdot/Sm_2_O_3_/Gd_2_O_3_ were 57.1 and 50.2 wt%, respectively. As a comparison, the calculated weight ratio of metal oxides in CNH/Fe_3_O_4_/Sm_2_O_3_/Gd_2_O_3_ and CNH/Fe_3_O_4_@Cdot/Sm_2_O_3_/Gd_2_O_3_ synthesized from 1.5 mg of Fe_3_O_4_, Sm_2_O_3_, Gd_2_O_3_, 0.135 mg of Cdot and 1.0 mg of CNH were 81.8 and 79.9 wt%, respectively. TGA results indicate that the practically loaded amounts of total metal oxides are lower than the expected amounts from charged raw materials due to the less reactivity of metal oxides on CNH.

Figure 3A shows the rough CNH surface, exhibiting the attachment of metal oxide NPs on CNH. Then, EDX analyses (Figure 3B) and elemental mapping images (Appendix A) of both CNH/Fe_3_O_4_/Sm_2_O_3_/Gd_2_O_3_ and CNH/Fe_3_O_4_@Cdot/Sm_2_O_3_/Gd_2_O_3_ contained elements of C, O, Fe, Sm, and Gd. Then weight ratios of Fe_3_O_4_:Sm_2_O_3_:Gd_2_O_3_ in CNH/Fe_3_O_4_/Sm_2_O_3_/Gd_2_O_3_ and CNH/Fe_3_O_4_@Cdot/Sm_2_O_3_/Gd_2_O_3_ calculated from EDX analysis were 1:0.036:0.073 and 1:0.012:0.092, respectively. The results indicate that the loaded amounts of Sm_2_O_3_ and Gd_2_O_3_ are lower than the estimated amounts form the synthesis procedure. Thus, the lower loaded amounts of total metal oxides than the expected amounts, which was estimated from TGA results, may results from the less reactivity of mainly Sm_2_O_3_ and Gd_2_O_3_ on CNH as estimated from EDX results.

When carboxylic acid in CNH was esterified with a terminal hydroxyl group of hydrophilic PEG chain, PEG@CNH kept the spherical shape. Still, the size of the TEM image increased from 96.7 ± 11.2 nm of pristine CNH to 128.1 ±12.0 nm (see Figure 4A(a,b)). PEG/CNH composite maintained the preferable dispersion property in water, and the hydrodynamic size changed from 117.7 nm of CNH to 176.5 nm because of the expansion of PEG by solvation in water. However, the zeta potential slightly decreased from −41.7 mV of CNH to −35.5 mV because of the coating by non-charged PEG. Even after coating by PEG, CNH/Fe_3_O_4_/Sm_2_O_3_/Gd_2_O_3_ and CNH/Fe_3_O_4_@Cdot/Sm_2_O_3_/Gd_2_O_3_ remain similar morphologies (Figure 4A(c,d)): their sizes were 129.1 ± 12.2 and 130.0 ± 14.8 nm, respectively, being almost similar sizes to pristine CNH after coating by PEG due to embedding of metal oxides in the void of CNH structure. The histograms of small particles on CNH were also plotted in Figure 4A(c,d). The obtained size distributions (11.0 ± 1.2 nm and 13.2 ± 2.1 nm, respectively) were consistent with those of Fe_3_O_4_ and Fe_3_O_4_@Cdot [30] but larger than those of Sm_2_O_3_ and Gd_2_O_3_ (4~5 nm) (see Figure 1C). Sm_2_O_3_ and Gd_2_O_3_ were not distinguished in TEM images due to their small sizes.

The hydrodynamic size in water compared among maters in Table 1. It increased compared to CNH and its complexes with single metal oxide, that is, hydrodynamic sizes (188.0 nm and 199.8 nm, respectively) of CNH/Fe_3_O_4_/Sm_2_O_3_/Gd_2_O_3_ and CNH/Fe_3_O_4_@Cdot/Sm_2_O_3_/Gd_2_O_3_ were almost 1.5-times compared to CNH, CNH/Sm_2_O_3_, CNH/Gd_2_O_3_, CNH/Fe_3_O_4,_ and CNH/Fe_3_O_4_@Cdot (117.7–135.7 nm). This finding was proportional to the size increase in nanocomposites as determined by TEM images.

However, the zeta potential, which reflects the surface charge of NPs to quantify the nanofluid stability [41], did not follow the tendency of particle sizes, as shown in Table 1. CNH possessed a larger negative zeta potential (−41.7 mV), but CNH/Sm_2_O_3_ and CNH/Gd_2_O_3_ had smaller potentials (−15.4 and −25.9 mV, respectively). The negatively charged surface of CNH is due to carboxyl groups on CNH. However, because of the esterification between carboxylic groups on CNH and hydroxylic groups on Sm_2_O_3_ and Gd_2_O_3_ NPs, CNH/Sm_2_O_3_ and CNH/Gd_2_O_3_ result in a decrease in the negative charge of CNH. It also appeared on CNH/Fe_3_O_4_ and CNH/Fe_3_O_4_@Cdot (−30.2 and −33.7 mV, respectively) due to the amidation reaction. Zeta potentials (−28.8 and −32.1 mV, respectively) of CNH/Fe_3_O_4_/Sm_2_O_3_/Gd_2_O_3_ and CNH/Fe_3_O_4_@Cdot/Sm_2_O_3_/Gd_2_O_3_ were similar to those (−27.60 and −30.79 mV, respectively) of PEG@CNH/Fe_3_O_4_/Sm_2_O_3_/Gd_2_O_3_ and PEG@CNH/Fe_3_O_4_@Cdot/Sm_2_O_3_/Gd_2_O_3_, maybe due to mainly binding of metal oxides on carboxyl groups of CNH [42]. However, these zeta potential values lead to greater interparticle repulsion. Hence, colloidal stability could be obtained [43].

Figure 4B shows time-course hydrodynamic sizes of CNH/Fe_3_O_4_/Sm_2_O_3_/Gd_2_O_3_ and CNH/Fe_3_O_4_@Cdot/Sm_2_O_3_/Gd_2_O_3_ before and after coating with PEG. The hydrodynamic size of these nanocomposites was slightly unstable in the first 24 h and then almost kept constant until 72 h. The hydrodynamic particle size of composites after PEG-coating was larger over a time period than before PEG-coating due to the expansion of PEG by solvation in water, and the particle size of PEG@CNH/Fe_3_O_4_@Cdot/Sm_2_O_3_/Gd_2_O_3_ was always larger than that of PEG@CNH/Fe_3_O_4_/Sm_2_O_3_/Gd_2_O_3_ because Cdot-coated Fe_3_O_4_ has a larger size than Fe_3_O_4_. Thus, Figure 4B demonstrates the time-independent stability of all nanocomposites and the systematic increase of hydrodynamic sizes by adding Cdot and PEG.

### 3.2. Drug Loading and Release of DOX and GEM 

Drug loading and release on nanocarriers (CNH, PEG@CNH, and PEG@CNH/Fe_3_O_4_@Cdot/Sm_2_O_3_/Gd_2_O_3_) are the most important parameter to evaluate the effectiveness of a drug delivery system [33]. GEM and DOX loaded on carriers are plotted as a function of drug concentrations in Figure 5A,B. Drug loading initially increased with dosage of the drug feeding and saturated at concentrations above 70 µg/mL (CNH and PEG@CNH) and above 90 µg/mL (PEG@CNH/Fe_3_O_4_@Cdot/Sm_2_O_3_/Gd_2_O_3_). The maximum loading capacities of GEM in CNH, PEG@CNH, and PEG@CNH/Fe_3_O_4_@Cdot/Sm_2_O_3_/Gd_2_O_3_ were 32.3%, 40.9%, and 55.4%, respectively, while the loading capacities of DOX were 43.4%, 59.1%, and 65.0%, respectively. PEG@CNH/Fe_3_O_4_@Cdot/Sm_2_O_3_/Gd_2_O_3_ with larger numbers of available binding sites cause higher loading capacity than CNH and PEG@CNH. Drugs bind on CNH and Cdot moieties possessing a graphitic domain and functional binding groups via π-π stacking, electrostatic, and hydrogen bonding interactions [44]. Moreover, metal oxides have a cavity and a protector molecule to host drug molecules [32].

Released GEM and DOX from carriers were investigated at pH 5.5 and 7.4 for tumor and physiological pH, respectively [45]. The release amounts of drugs increased rapidly till 24 h and converged (Figure 5C–E). As compared in Figure 5F, amounts released after 72 h were almost consistent between DOX and GEM, being almost independent of carriers, but the release at the tumor environment (pH 5.5) was about 2-fold higher than at physiological pH (7.4). This pH dependence of drug release is preferable as the activity of drugs on the tumor therapy since a higher release was obtained at pH 5.5 rather than at pH 7.4. At acidic conditions (pH 5.5), the protonation of COO^−^ groups on CNH breaks the electrostatic interaction with ammonium ion groups on DOX and GEM, leading to the release of DOX and GEM from carriers [46].

### 3.3. Co-Loading and Release of Drugs

Co-delivery of GEM and DOX on a single carrier (CNH and PEG@CNH) was performed to enhance cancer treatment. Co-loading of drugs at an equal initial weight amount on a carrier was plotted as a function of each drug concentration in Figure 6A. Although loading amounts of both drugs were saturated at concentrations above 70 µg/mL, but the encapsulated amount of DOX (0.31 mg/mg(carrier)) was better than GEM (0.19 mg/mg(carrier)). Meanwhile, amounts of DOX (0.41 mg/mg(carrier)) and GEM (0.25 mg/mg(carrier)) loaded on PEG@CNH were higher than on CNH. Both drugs can be simultaneously loaded on carriers with *π*-*π* stacking, electrostatic interaction, and hydrogen bonding [44]. Then, the difference in loading amounts depends on the hydrophilicity and molecular size of drugs, the number of binding sites on carriers, and the affinity of drugs with carrier components. The loading of DOX higher than GEM is due to the strong *π*-*π* stacking and hydrogen bonding of DOX with carriers. This tendency is similar to the independent loading of both drugs on CNH and PEG@CNH (Figure 5A). The better drug loading on PEG@CNH than CNH is the contribution of binding sites of PEG [47].

Co-release of GEM/DOX on carriers was also investigated at pH 5.5 and 7.4. As presented in Figure 6B, the release took 12–72 h, depending on the carrier. The release amounts of GEM and DOX after 72 h from CNH and PEG@CNH were similar at pH 5.5 and 7.4. However, the co-release of drugs in a tumor environment (pH 5.5) was almost half of that in a physiological environment (pH 7.4). These release behaviors and amounts are consistent with the release of the independent drug (Figure 5F). Thus, the results indicate that co-loaded GEM and DOX independently release without interference with each other, different from the loading behavior. Thus, the co-delivery system plays better anticancer efficacy than the delivery of GEM and DOX alone.

Additionally, a different combination of cancer drugs was targeted for co-loading/release. Hydrophilic GEM composed of hydroxyl, carboxylic acid, and amine groups is water-soluble. However, hydrophobic CPT possessing hydroxyl, aromatic ring, and carbonyl groups is not soluble in water but soluble in methanol. Thus, drug loading on carriers was carried out in the water for GEM and in methanol for CPT. Amounts of drug loading on CNH, CNH/Fe_3_O_4_, and CNH/Fe_3_O_4_@Cdot are plotted in Figure 7. GEM@CNH/Fe_3_O_4_ and CNH/Fe_3_O_4_@Cdot were saturated at 0.13 mg/mL, with a loading capacity of 0.173 and 0.103 mg/mg(carrier), respectively. However, GEM-loading on CNH increased even above 0.13 mg/mL. The loading amount of CPT on CNH/Fe_3_O_4_@Cdot, CNH/Fe_3_O_4_, and CNH displayed similar behaviors to the GEM-loading. However, the saturation concentration (0.19 mg/mL) of CPT was higher, and the saturated CPT amounts on CNH/Fe_3_O_4_ (0.261 mg/mg(carrier)) and CNH/Fe_3_O_4_@Cdot (0.117 mg/mg(carrier)) were also higher than those of GEM. Similarly, the CPT@CNH continued to increase. It might be due to the strong π-π stacking attraction of the aromatic ring of CPT with the graphitic structure of CNH and the strong hydrogen bonding of hydroxyl group of CPT with carboxylic acid group of CNH. As GEM possesses less aromatic ring contrasting to CPT, the hydrogen bonding or electrostatic interaction between GEM and CNH is more dominant than the π-π stacking interaction. Bare CNH can adsorb many drugs on its binding sites, but the attachment of Fe_3_O_4_ on CNH loses parts of binding sites, and the binding of Fe_3_O_4_@Cdots accelerates the loss of the binding sites of CNH.

Co-loading of GEM and CPT on the carrier was also performed using a carrier half-saturated by CPT and loading with GEM. As seen in Figure 7C, the saturated loading amount of GEM was 0.047 mg/mg(carrier) on CNH/Fe_3_O_4_@Cdot and 0.085 mg/mg(carrier) on CNH/Fe_3_O_4_. These values ideally match GEM loading amounts on carriers half-saturated by CPT (0.051 and 0.086 mg/mg(carrier), respectively), indicating that GEM occupies the remaining non-CPT-loaded binding site on carriers. These loading behaviors are useful for chemotherapies using multiple drugs and simultaneously for hyperthermia by carriers, including iron oxide NP or iron oxide NP-coated carbon dots.

The release of co-loaded cancer drugs from CNH/Fe_3_O_4_ and CNH/Fe_3_O_4_@Cdot carriers was also examined under different pH (5.4 and 7.4), temperature (37 °C and 42 °C), and under blue LED light irradiation, as shown in Figure 8. Since the release of water-insoluble hydrophobic CPT from the carrier was not observed because absorption bands of CPT were not detected in the UV-visible absorption spectrum of supernatant in a buffer solution, the release was calculated only for GEM.

The release of GEM was carried out from GEM/CPT@CNH/Fe_3_O_4_ at 37 °C, and the temperature was increased to 42 °C after 24 h. As shown in Figure 8A, the released-GEM rose during the initial 5 h at both pH 5.4 and pH 7.4 (56% and 44%, respectively), remaining the same values until 24 h. The release in the second period at 42 °C increased by 13–14% after 2 h. Incidentally, the released-GEM from GEM/CPT@CNH/Fe_3_O_4_ at 42 °C (Figure 8B) reached 64% (at pH 5.4) and 54% (at pH 7.4) after 5 h. These values were close to those of two-step release (69 and 58%, respectively), indicating that the increased temperature is an important factor in the drug release [48].

The release of GEM from GEM/CPT@CNH/Fe_3_O_4_@Cdot was also carried out with blue LED irradiation. After GEM was released at the body temperature for 24 h, blue LED light was irradiated (6 h) and caused an additional release of around 5–7% due to the photothermal effect of Cdot, as shown in Figure 8C. As a comparison, the released-GEM from GEM@CNH/Fe_3_O_4_@Cdot at pH 5.4 and 7.4 (37 °C) was about 40% and 32%, respectively, and increased approximately 5–6% after 2 h light irradiation. These increases are consistent with the case of GEM/CPT@CNH/Fe_3_O_4_@Cdots. However, it should be noted that the released-GEM from GEM@CNH/Fe_3_O_4_@Cdot was almost half of the release from GEM/CPT@CNH/Fe_3_O_4_@Cdots, meaning that half of the total drugs released, that is, half of total GEM in the former carrier and GEM in the latter carrier.

The release of GEM from GEM/CPT@CNH/Fe_3_O_4_@Cdots was also carried out at cell death temperature (42 °C), as shown in Figure 8D. The released-GEM at pH 5.4 and 7.4 were 92% and 80%, respectively, after 5 h. These values were higher than that of GEM/CPT@CNH/Fe_3_O_4_@Cdot using irradiation light. The results mention that the increased temperature using the hyperthermia effect is more effective for GEM release than the photothermal effect of Cdot by LED irradiation.

The results of release at different conditions are plotted in Figure 8E. As a result, the release of GEM is always better at pH 5.4 than at pH 7.4, and the co-loading of GEM and CPT shows that GEM can be released more abundantly than at a single loading system of GEM. It can be assumed that the drug will be released more easily from fully occupied binding sites. Moreover, the release of GEM at a temperature of cell death (42 °C) is preferable to that at human body temperature (37 °C). Additionally, Cdot possesses a photothermal efficiency [9]. Thus, carriers, including Cdot, also increase the drug release. A double loading system results are preferable for cancer therapy because of larger GEM release. CNH/Fe_3_O_4_@Cdot is more effective on drug release than CNH/Fe_3_O_4_, and the former has efficacy in the photodynamic/photothermal effects. Thus, the Cdot-including material is a preferable delivery system.

### 3.4. In Vitro Cytotoxicity

The cytotoxic activity of CNH/Fe_3_O_4_@Cdot, CPT@CNH/Fe_3_O_4_@Cdot, and GEM&CPT@CNH/Fe_3_O_4_@Cdot in NCI-N87 cells was examined using a crystal violet assay, which is a simple method for investigating cell survival and growth inhibition by staining attached cells with crystal violet dye. Dead cells lose their adherence and are thus removed from the cell population, lowering the number of crystal violet staining in a culture [49]. IC_50_ values (half of the maximum inhibitory concentration) were employed to determine the degree of cytotoxicity. While CNH/Fe_3_O_4_@Cdot nanocarrier as a control showed an IC_50_ value higher than 1.6 mg/mL IC_50_ of CPT@CNH/Fe_3_O_4_@Cdot and GEM&CPT@CNH/Fe_3_O_4_@Cdot was 1.3 mg/mL, demonstrating the increased cytotoxicity (Figure 9). These results indicate that drugs loading on CNH/Fe_3_O_4_@Cdot nanocarrier contributes to cytotoxicity for NCI-N87 cells. Thus, the loaded drugs on the present carrier are potentially used in cancer chemotherapy.

## 4. Conclusions

Both CNH/Fe_3_O_4_/Sm_2_O_3_/Gd_2_O_3_ and CNH/Fe_3_O_4_@Cdot/Sm_2_O_3_/Gd_2_O_3_ were successfully synthesized and performed covalent bonding with PEG. NPs were well-dispersed on CNH without changing the morphology of CNH, and the hydrodynamic size of nanocomposite exhibited time-independent stability. Due to various available binding sites, the drug loading of nanocomposites is higher than pristine CNH. These nanocomposites showed high drug loading capacity and drug release profile, selectively, in tumor conditions (pH 5.5), where the release was about 2-fold higher than at physiological pH (7.4), resulting in drug release enhancement from the carrier and destruction of cancer cells at acidic condition. The drug release in acidic conditions is because the protonation of COO^−^ groups on CNH breaks the electrostatic interaction with ammonium ion groups on drugs. Moreover, the principle of hyperthermia is thermogenesis by Fe_3_O_4_ NP under the magnetic field, and the thermal effect by Fe_3_O_4_ stimulates the release of anticancer drugs, which enhances the efficacy of chemotherapy similarly for CNH/Fe_3_O_4_ and CNH/Fe_3_O_4_@Cdot [30]. The superior photothermal activity by Cdot compared to Fe_3_O_4_@Cdot gives it the advantage on drug release and cancer cell destruction to CNH/Fe_3_O_4_ without Cdot, since Cdot adsorbs the irradiated light energy and can transfer it to heat and the heat affects the breakdown of physical interaction between carrier materials and drugs. In addition, the co-delivery system of anticancer drugs performed better anticancer efficacy than independent drug delivery. Such stimulation (pH change, temperature rise and magnetic force) could give rise to the break of physical interactions (the π-π interaction, the hydrogen bonding and the electrostatic attraction) of drugs with carriers. Moreover, the simultaneous stimulation by multi-drugs, pH, heat and magnetic force will act most effectively for the butchery of cancer cells. Thus, PEG-coated CNH containing samarium, gadolinium, and iron oxides, or C-dot coated iron oxide NPs possesses potential as nanocarriers for anticancer drugs and theranostic (combining therapy and diagnostics) materials, which are suitable for developing advanced medication for combination therapy.

## Data Availability

Not applicable.

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
