# Peer review of "Preparation of Functional Nanoparticles-Loaded Magnetic Carbon Nanohorn Nanocomposites towards Composite Treatment"

_nanomaterials, 2023, doi:10.3390/nano13050839_

Round 1
Reviewer 1 Report
The current work focuses on the Preparation of Functional Nanoparticles-Loaded Magnetic Carbon Nanohorn Nanocomposites for Composite Treatment. The author’s some effort into the improved manuscript but major issues should be addressed.
Major issues
-Abstract, multitherapy application has been intensively studied. So, first, show the novelty of this work and then the main outcomes.
- The introduction doesn’t provide sufficient background, and the most relevant references are not included. In addition, the novelty of this work is not highlighted and the author's contribution was unclear compared to other previous works.
- One of the main problems in the manuscript is that the authors only show results without interpretations or confirmation by citation. More details are required to explain the obtained results.
Minor issues
-Materials and Methods, where the source and procuresses for preparation Fe3O4?
-Schemes 1. Should be supported by the chemically bond formation and the chemical reaction to clear it to the reader
- Results and discussion, why you claiming the prepared were magnetite (Fe3O4)? No analysis in the manuscript confirms or supports this claim. It may be maghemite or mixed iron oxide
-IR absorption spectra, where is the characterized peak for Fe3O4 in the composites? And where is the characterized peak for the amide bond in the matrix?
-Live/dead fluorescence imaging of the cultured cell is required to clear Viability to the reader
-Hyperthermia effect by Fe3O4 was not focused and highlighted
-Line 174, correct typo CH2
-Line 297, correct citation style
Author Response
Attached a file.

Reviewer 2 Report
Nanocomposite technology for drug delivery and treatment is a key technology in Theranostics. This article was of great interest to this reviewer as it deals with the emerging topic of nano-therapy. The concept of this study looks good, and the authors put a lot of effort into it. However, this reviewer believes that this manuscript needs some improvement.
1. The intro part needs improvement.
The authors can highlight the originality of the research by further emphasizing the advantages and improvements related to research using multimodal nanoparticles. It is also recommended to receive professional proofreading so that good research contents can be highlighted.
2. It seems good to place some data related to nanomaterial synthesis results as supplementary data.
It is true that all the data in the figure is important. However, it would be good to place some data as supplementary material to focus more on the interpretation related to the nanocomposite's operational performance and in-vitro experimental results.
In addition, if this paper goes through minor supplementary work such as figure formatting, this paper will be accepted.
Author Response
Attached a file.

Reviewer 3 Report
This manuscript has described synthesis of two nanocomposites and examined their use for drug delivery in order to treat cancer, but questions arise for the reader:
- While Fe3O4 can also play a role as an MRI contract reagent, so Gd2O3 can be omitted. What is the need to use Gd2O3?
- Clarify for the reader why you compared these two nanocomposites? What features did you expect to give to your composite by carbon dot?
- The quality of figure 7 is very poor.
- 2.5. Viability test and 3.4. In vitro cytotoxicity: the number of repetitions is not mentioned.
- 2.5. Viability test and 3.4. In vitro cytotoxicity: you investigate the use of PEG@CNH/Fe3O4/Sm2O3/Gd2O3 and PEG@CNH/Fe3O4@Cdot/Sm2O3/Gd2O3, but you study the cytotoxicity of CNH/Fe3O4@Cdot. Why did not you consider these tow target nanocomposites? Please clarify it for readers.
- In figure 9, The standard deviation is very high and it seems that there is no significant difference between any of the data specially for CPT.
Author Response
Attached a file.

Reviewer 4 Report
Manuscript has a title Preparation of Functional Nanoparticles-Loaded Magnetic Carbon Nanohorn Nanocomposites towards Composite Treatment. The manuscript is devoted to the method of synthesis of composite particles, which are graphene particles acting as a matrix, which is filled with nanoparticles of magnetite and oxides of gadalinium and samarium. The particles can adsorb anticancer drugs on themselves. Methods for the synthesis of composite nanostructured drug carriers, combinations of nanostructures in composites, interaction of drugs with the obtained carriers, loading and release under various conditions (temperature, pH, composition), as well as the toxicity of the obtained therapeutic agents were studied.
The manuscript contains a large amount of multidisciplinary data, is well structured, clearly and logically presented material, corresponds to the subject of the journal and can be published in the journal with minor modifications.
1. The literature review is sufficient, however, it is necessary to consider in more detail the requirements and examples of nanoplatforms for theranostics and identify ways to use them. For example, Korolkov I.V., Zibert A.V., Lissovskaya L.I., Ludzik K., Anisovich M., Kozlovskiy A.L., Shumskaya A.E., Vasilyeva M., Shlimas D.I., Jażdżewska M., Marciniak B., Kontek R., Chudoba D., Zdorovets M.V. Boron and Gadolinium Loaded Fe3O4 Nanocarriers for Potential Application in Neutron Capture Therapy// International Journal of Molecular Sciences. – 2021 – Vol. 22(16). – P.8687. https://doi.org/10.3390/ijms22168687
2. Emphasize the novelty of your work in comparison with previous works of other authors.
3. One of the important aspects of the considered platforms is the magnetic properties of the composite at various stages of formation. If possible, please provide information about your platforms.
4. The influence of the magnetic field on the change in the temperature of the resulting nanostructures, which is important for hyperthermia, the area in which the use of the resulting platforms is envisaged, has not been considered.
5. The EDX method does not allow one to adequately estimate the composition of a bulk composite, especially if metal oxide particles are embedded with graphene particles. Perhaps more informative would be XRD for the composites considered in Figure 3 and give the phase composition of the composite. Mapping is almost indistinguishable, and in the case of a heterogeneous composite, it is not of great interest. Accordingly, you can transfer these drawings to the templates.
6. Figure 4c and 4c are best presented in the form of a table.
7. Improve the image in Figure 7.
8. Simulation of irradiation of composites to increase the release of drugs has shown effectiveness. What is it connected with? In addition, it is not clear how this will work in a live experiment (give an example).
9. Why are drug loading and release data significantly different for different pH?
10. Despite the good results in the co-adsorption of two drugs on carrier platforms, their relative effectiveness will be clearer if converted to milligrams of the drug itself, and not the composite as a whole.
11. Expand, please, conclusions, having presented comparative numerical data.

Author Response
Attached a file.

Round 2
Reviewer 1 Report
Accept in the present form
Author Response
Thank you for your support.
Reviewer 3 Report
After careful evaluation the responses of the respected authors, it seems that all my suggestions for the correction of the article have been taken into consideration, and the publication of the article in its current form is appropriate in my opinion.
Author Response
Thank you for your suggestion.